# Adjuvant BRAF-MEK Inhibitors versus Anti PD-1 Therapy in Stage III Melanoma: A Propensity-Matched Outcome Analysis

**DOI:** 10.3390/cancers15020409

**Published:** 2023-01-07

**Authors:** Melissa M. De Meza, Willeke A. M. Blokx, Johannes J. Bonenkamp, Christian U. Blank, Maureen J. B. Aarts, Franchette W. P. J. van den Berkmortel, Marye J. Boers-Sonderen, Jan Willem B. De Groot, John B. A. G. Haanen, Geke A. P. Hospers, Ellen Kapiteijn, Olivier J. Van Not, Djura Piersma, Rozemarijn S. Van Rijn, Marion Stevense-den Boer, Astrid A. M. Van der Veldt, Gerard Vreugdenhil, Alfonsus J. M. Van den Eertwegh, Karijn P. M. Suijkerbuijk, Michel W. J. M. Wouters

**Affiliations:** 1Scientific Bureau, Dutch Institute for Clinical Auditing, Rijnsburgerweg 10, 2333AA Leiden, The Netherlands; 2Department of Biomedical Data Sciences, Leiden University Medical Center, Albinusdreef, 2333ZA Leiden, The Netherlands; 3Division of Surgical Oncology, Netherlands Cancer Institute, Plesmanlaan 121, 1066CX Amsterdam, The Netherlands; 4Department of Pathology, University Medical Center Utrecht, Heidelberglaan 100, 3584CX Utrecht, The Netherlands; 5Department of Surgery, Radboud University Medical Center, Geert Grooteplein Zuid 10, 6525GA Nijmegen, The Netherlands; 6Division of Molecular Oncology and Immunology, Netherlands Cancer Institute, Plesmanlaan 121, 1066CX Amsterdam, The Netherlands; 7Division of Medical Oncology, Netherlands Cancer Institute, Plesmanlaan 121, 1066CX Amsterdam, The Netherlands; 8Department of Medical Oncology, Leiden University Medical Center, Albinusdreef 2, 2333ZA Leiden, The Netherlands; 9Department of Medical Oncology, GROW-School for Oncology and Reproduction, Maastricht University Medical Center, P. Debyelaan 25, 6229HX Maastricht, The Netherlands; 10Department of Medical Oncology, Zuyderland Medical Center Sittard, Dr. H. van der Hoffplein 1, 6162BG Sittard-Geleen, The Netherlands; 11Department of Medical Oncology, Radboud University Medical Center, Geert Grooteplein Zuid 10, 6525GA Nijmegen, The Netherlands; 12Isala Oncology Center, Dokter van Heesweg 2, 8025AB Zwolle, The Netherlands; 13Department of Medical Oncology, University Medical Center Groningen, Hanzeplein 1, 9713GZ Groningen, The Netherlands; 14Department of Medical Oncology, University Medical Center Utrecht, Heidelberglaan 100, 3584CX Utrecht, The Netherlands; 15Department of Internal Medicine, Medisch Spectrum Twente, Koningsplein 1, 7512KZ Enschede, The Netherlands; 16Department of Internal Medicine, Medical Center Leeuwarden, Henri Dunantweg 2, 8934AD Leeuwarden, The Netherlands; 17Department of Internal Medicine, Amphia Hospital, Molengracht 21, 4818CK Breda, The Netherlands; 18Departments of Medical Oncology and Radiology & Nuclear Medicine, Erasmus Medical Center, ‘s-Gravendijkwal 230, 3015CE Rotterdam, The Netherlands; 19Department of Internal Medicine, Maxima Medical Center, De Run 4600, 5504DB Eindhoven, The Netherlands; 20Department of Medical Oncology, Amsterdam UMC, VU University Medical Center, Cancer Center Amsterdam, De Boelelaan 1118, 1081HZ Amsterdam, The Netherlands

**Keywords:** melanoma, adjuvant therapy, targeted therapy, immune checkpoint inhibition

## Abstract

**Simple Summary:**

BRAF/MEK therapy and anti-PD-1 therapy have shown better recurrence-free survival of stage III melanoma in patients with BRAF V600 mutations in clinical trials. However, little is known about how these therapies compare to each other in everyday practice. The aim of our study was to describe the toxicity and survival of patients treated with BRAF/MEK therapy and anti-PD-1 therapy in daily practice. We demonstrated that grade ≥ 3 toxicity occurred in 11.5% of patients and was the most common cause of early treatment discontinuation (71.1%). We also show that at 12 months, patients treated with BRAF/MEK therapy have less progression than those treated with anti-PD-1 therapy. However, this is no longer the case at 18 months.

**Abstract:**

Adjuvant BRAF/MEK- and anti-PD-1 inhibition have significantly improved recurrence-free survival (RFS) compared to placebo in resected stage III BRAF-mutant melanoma. However, data beyond the clinical trial setting are limited. This study describes the toxicity and survival of patients treated with adjuvant BRAF/MEK inhibitors and compares outcomes to adjuvant anti-PD-1. For this study, stage III BRAF V600 mutant cutaneous melanoma patients treated with adjuvant BRAF/MEK-inhibition or anti-PD-1 were identified from the Dutch Melanoma Treatment Registry. BRAF/MEK- and anti-PD-1-treated patients were matched based on propensity scores, and RFS at 12 and 18 months were estimated. Between 1 July 2018 and 31 December 2021, 717 patients were identified. Of these, 114 patients with complete records were treated with BRAF/MEK therapy and 532 with anti-PD-1. Comorbidities (*p* = 0.04) and geographical region (*p* < 0.01) were associated with treatment choice. In 45.6% of BRAF/MEK-treated patients, treatment was prematurely discontinued. Grade ≥ 3 toxicity occurred in 11.5% of patients and was the most common cause of early discontinuation (71.1%). At 12 and 18 months, RFS in BRAF/MEK-treated patients was 85% and 70%, compared to 68% and 68% in matched anti-PD-1-treated patients (*p* = 0.03). In conclusion, comorbidities and geographical region determine the choice of adjuvant treatment in patients with resected stage III BRAF-mutant melanoma. With the currently limited follow-up, BRAF/MEK-treated patients have better RFS at 12 months than matched anti-PD-1-treated patients, but this difference is no longer observed at 18 months. Therefore, longer follow-up data are necessary to estimate long-term effectiveness.

## 1. Introduction

The phase 3, double-blind, placebo-controlled COMBI-AD trial demonstrated a significantly longer recurrence-free survival (RFS) after 12 months for adjuvant dabrafenib plus trametinib compared to placebo in patients with resected stage III melanoma with BRAF V600E or V600K mutations [1]. In 2020, updated data showed continued RFS benefit and distant-metastasis-free survival (DMFS) compared to placebo, without significant long-term toxic effects [2]. Subsequently, adjuvant BRAF/MEK-inhibition therapy was approved and reimbursed for the treatment of resected stage III melanoma patients in the Netherlands in November 2020. In contrast, adjuvant anti-PD-1 treatment was approved and reimbursed in December 2018 [3,4]. Since their approval and reimbursement, adjuvant anti-PD-1 or BRAF/MEK inhibitors have been considered the standard care treatment in resected stage III melanoma patients in the Netherlands [5,6,7].

A head-to-head comparison of adjuvant BRAF/MEK- and adjuvant anti-PD-1-treated patients has not been made in clinical trials. To our knowledge, there are also no ongoing trials comparing targeted therapy and immune-checkpoint inhibition at the time of writing this report. Data from population-based real-world cohorts, such as the Dutch Melanoma Treatment Registry (DMTR)., could be used to directly compare the effectiveness of adjuvant BRAF/MEK-inhibition to that of adjuvant anti-PD-1 treatment in daily practice. However, such a comparison could also be hampered by indication bias. Propensity score matching (PSM) can partly correct for such biases and form an alternative to a randomized controlled trial (RCT).

Propensity score matching is a statistical method that can be used to reduce confounding effects and make an unbiased estimate of a treatment effect [8]. Similar to randomization in RCTs, PSM creates two even groups with a similar distribution of baseline characteristics between treatment groups, removing confounding effects when comparing outcomes [8,9,10]. In PSM, patients are matched based on their propensity scores, which is the chance of receiving a treatment based on measured confounders (age, tumor stage, etc.). PSM allows for estimating the difference in outcomes between two groups that are identical in all aspects, except that the treatment regimens are different for the two groups (marginal treatment effect) [8].

Resectable stage III melanoma patients generally have a better prognostic profile compared to unresectable stage III/IV melanoma patients and have a considerable chance to remain tumor-free after complete surgical resection. However, the considerations in choosing BRAF/MEK or anti-PD-1 as first-line therapy in the advanced melanoma setting (i.e., relatively quick anti-tumor effect of BRAK/MEK inhibitors versus less acquired resistance in case of immunotherapy) can be different in the adjuvant setting. Therefore, one could hypothesize that first-line adjuvant treatment preference is rather based on specific patient preferences and less on tumor characteristics, without a difference in the outcome of patients treated with adjuvant BRAF/MEK-inhibition versus anti-PD-1 treatment. In this study, we describe the baseline characteristics, toxicity, and survival rates of patients treated with BRAF/MEK-inhibition therapy in a nationwide Dutch cohort. Using PSM, we also compare survival rates of BRAF/MEK-treated patients to those of adjuvant anti-PD-1-treated patients beyond the clinical trial setting.

## 2. Materials and Methods

### 2.1. Study Population

The patients included in this study were identified using the Dutch Melanoma Treatment Registry. The DMTR is a prospective nationwide registry containing data on all unresectable stage III/IV melanoma patients and resectable stage III/IV patients treated with (adjuvant) systemic therapy since 2012 and 2018, respectively [11,12]. BRAF-mutant patients with American Joint Committee on Cancer (AJCC) 8th edition [13] resectable stage III cutaneous melanoma treated with adjuvant BRAF/MEK-inhibition therapy or anti-PD-1 registered between 1 July 2018 and December 31st 2021 were included in this study. The data cut-off was 4 April 2022.

In the DMTR, data are registered by trained data managers and approved by medical oncologists representing the 14 melanoma centers in the Netherlands. The medical ethical committee approved research using DMTR data, and research with DMTR data were not deemed subject to the Medical Research Involving Human Subjects Act in compliance with Dutch regulations.

### 2.2. Statistical Analysis

Descriptive statistics were used to describe patient and tumor characteristics. A multivariable logistic regression analysis identified factors associated with receiving BRAF/MEK-inhibition therapy versus anti-PD-1 treatment as first-line adjuvant therapy. A multivariable logistic regression analysis was also used to estimate propensity scores for BRAF/MEK-inhibitor- and anti-PD-1-treated patients. Melanoma centers were grouped into northern, southern, or middle regions based on their geographical position.

Toxicity rates for adjuvant BRAF/MEK-inhibition therapy were illustrated using descriptive statistics. Toxicity was graded using the Common Terminology Criteria for Adverse Events (CTCAE) version 5.0. Only CTCAE grade ≥ 3 treatment-related toxicity and any grade toxicity necessitating treatment discontinuation are registered in the DMTR. Toxicity rates for adjuvant anti-PD-1-treated patients were described before [12].

Premature treatment discontinuation rates and reasons for premature discontinuation of BRAF/MEK-inhibition therapy were described. Treatment discontinuation before 12 months was considered premature discontinuation. Premature discontinuation rates were illustrated in a stacked bar chart. Adjuvant BRAF/MEK-inhibition therapy duration was depicted as the time from starting therapy to the last prescription date. As most prescriptions were written for a month, patients receiving their last prescription at 10.5 months or later were considered to have been treated for a full year.

### 2.3. Propensity Score Matching

BRAF mutant patients treated with adjuvant anti-PD-1 therapy were matched to patients treated with adjuvant BRAF/MEK-inhibition using propensity score matching. Patients were matched according to the nearest neighbor with the caliper matching method. Patients were matched for the following items: age at diagnosis (<65 years versus ≥65 years), sex, the presence of comorbidities, ECOG performance score, and AJCC8 disease stage. Covariates used for matching were chosen based on clinical expertise and the previous literature identifying these factors as prognostic factors [8,14,15]. Only patients with complete records based on the above-mentioned items were included in the statistical analysis included in this paper. Patients were matched using a ratio of 1:1 (BRAF/MEK versus anti-PD-1-treated patients) and were matched with patients with propensity scores within a pre-set caliper. Standardized mean differences (SMD) were calculated for all covariates to assess the matching quality. Standardized mean differences of <0.1 were considered negligible. We also performed a second analysis using optimal matching to address potential biases in the nearest neighbor caliper matching procedure. Optimal matching minimizes the average within-pair difference in propensity scores when matching patients [16]. Because matched patients had baseline characteristics that were more similar than randomly selected subjects, a comparison of RFS and OS between matched patients was performed using a stratified log-rank test [8]. A univariable cox proportional hazards model with a robust variance estimator was used to estimate the relative change in the hazard ratio (HR) of survival between matched patient cohorts [17].

Recurrence-free survival (RFS) rates at 12 and 18 months and overall survival (OS) rates at 12 and 18 months were estimated for patients treated with adjuvant BRAF/MEK-inhibition and anti-PD-1 therapy. The median follow-up duration was calculated using the reversed Kaplan–Meier method. RFS was calculated from the start of systemic therapy until recurrence or death. OS was calculated from the start of systemic therapy until death. Patients who did not meet the endpoints for RFS or OS were censored at the date of the last follow-up.

Data handling and statistical analyses were performed using the R software system for statistical computing (version 4.2.1.; packages ggplot2, coxrobust, coxphw, plyr, magrittr, RColorBrewer, EnvStats, lubridate, MatchIt, cmprsk, dplyr, forestmodel, survminer, tableone, survival, tidyverse, stringr, tidyr, readxl) [18,19,20,21,22,23,24,25,26,27,28,29,30,31,32,33,34,35,36].

## 3. Results

### 3.1. Patient and Tumor Characteristics

Between 1 July 2018 and 31 December 2021, 717 resectable stage III BRAF-mutant cutaneous melanoma patients were included in the DMTR (Figure 1). We identified 532 complete cases of patients 18 years or older treated with anti-PD-1 therapy and 114 complete cases of patients treated with BRAF/MEK inhibitors (Table 1). All patients treated with adjuvant BRAF/MEK-inhibition therapy received dabrafenib/trametinib. Age, sex, ECOG PS, and disease stage did not differ between BRAK/MEK-inhibitor- and anti-PD-1-treated patients. Of the BRAF/MEK-inhibitor-treated patients, 78.1% had a BRAF-V600E mutation, 17.5% had a BRAF non-V600E mutation (11 (9.6%) had a V600K mutation, 1 patient (0.9%) had a V600R mutation, and 8 (7.0%) patients were registered as “other BRAF mutation”), and in 4.4% the BRAF mutation type was registered as unknown, compared to 72.6%, 22.0%, and 5.5% in anti-PD-1-treated patients, respectively (*p* = 0.48) (Appendix A). BRAF/MEK-inhibitor-treated patients significantly more often had comorbidities than anti-PD-1-treated patients (74.6% versus 63.5%, *p* = 0.03). Other patient and tumor characteristics are described in Appendix A. BRAF/MEK-inhibitor-treated patients more often had autoimmune and musculoskeletal comorbidities (20.2% versus 3.4% (*p* < 0.01) and 19.3% versus 8.6% (*p* < 0.01), respectively) and more often had undergone organ transplants (2.6% versus 0%, *p* < 0.01) compared to anti-PD-1 patients (Appendix A). BRAF/MEK-treated patients also had more co-medication at diagnosis than anti-PD-1-treated patients. (Appendix A).

### 3.2. Predictors for Receiving BRAF-MEK Inhibitors

In stage III BRAF-mutant patients, comorbidities (OR = 1.67, *p* = 0.04) and northern geographical region (OR = 3.05, *p* < 0.01) were associated with adjuvant BRAF/MEK-inhibition therapy over anti-PD-1 treatment. Age, sex, AJCC stage, and ECOG PS were not significantly associated with receiving BRAF/MEK inhibitors (Table 2).

### 3.3. Toxicity Rates of Adjuvant BRAF/MEK-Inhibition Therapy in Daily Clinical Practice

Of the current 114 patients treated with adjuvant BRAF/MEK-inhibition therapy, toxicity data were available for 104 (91.2%) patients. Twelve (11.5%) patients experienced one or more grade ≥3 adverse events (Table 3). The most common grade ≥3 adverse events in these patients were pyrexia (4.8%) and skin toxicities (3.8%).

We previously described 18.3% of adjuvant anti-PD-1-treated patients developing grade ≥3 toxicity [12]. The most common toxicities were colitis/diarrhea (4.6%), hepatitis (1.1%), rash/pruritus (0.5%), dyspnea/pneumonitis (1.1%), and “other” in 6.8%.

### 3.4. Premature Discontinuation of Adjuvant BRAF/MEK-Inhibition Therapy in Daily Clinical Practice

Of the current 114 BRAF/MEK-inhibitor-treated patients, 75 discontinued or completed treatment during this study’s follow-up period, while 39 patients were still undergoing BRAF/MEK-inhibition treatment. Fifty-two (45.6%) patients prematurely discontinued treatment (Figure 2). Thirty-seven (32.5%) of the 114 BRAF/MEK-treated patients prematurely discontinued treatment due to any grade toxicity. Of these 37 patients, second-line treatment was registered for 11 patients. Seven patients received second-line anti-PD-1 therapy after early BRAF/MEK-inhibition discontinuation due to any grade toxicity, one patient received BRAF/MEK therapy, one receive combination ipilimumab and nivolumab, one underwent surgery only, and one underwent surgery and radiotherapy. These patients had not developed progression before BRAF/MEK discontinuation. Reasons for treatment discontinuation in discontinued patients (n = 52) were toxicity (71.1%), patient’s choice (13.5%), poor patient condition (7.7%), progression (3.8%), and other (3.8%). We previously described a premature discontinuation rate of 61.0% in adjuvant anti-PD-1-treated patients [12].

### 3.5. Recurrence-Free Survival (RFS) and Overall Survival (OS)

In the original cohort RFS in BRAF/MEK-inhibitor-treated and anti-PD-1-treated patients at 12 and 18 months were 84.6% (95% CI, 76.8–93.3) and 67.6% (95% CI, 54.9–83.1), compared to 70.1% (95% CI, 65.9–74.5) and 65.4% (95% CI, 60.7–70.4), respectively (*p* = 0.06, log-rank test) (Appendix A). Median RFS was 30.1 months (95% CI, 20.8-NR) in patients receiving adjuvant BRAF/MEK inhibitors versus 35.5 months (95% CI, 35.5-NR) in patients receiving adjuvant anti-PD-1. The median follow-up time in BRAF/MEK-inhibitor-treated patients was 10.8 months (IQR, 4.6–17.8) and 13.9 months (IQR 7.3–20.5) in anti-PD-1-treated patients.

Twenty-one patients developed recurrence in the original BRAF/MEK-inhibitor-treated cohort during the follow-up period of this study, while one patient died without having developed recurrence. Of these patients, 14 (63.6%) had a subsequent treatment registered in the DMTR database. Of the patients with subsequent treatment, one patient (7.1%) received surgery alone, three (21.4%) received systemic therapy alone, two (14.3%) received systemic therapy in combination with surgery, four (28.6%) received radiotherapy alone or in combination with other treatment, two (14.3%) patients received T-VEC, and two (14.3%) patients were registered as receiving other treatment (Appendix A). Six patients in the original BRAF/MEK-inhibitor-treated cohort died during the follow-up of this study.

Overall survival (OS) in BRAF/MEK- and anti-PD-1-treated patients at 12 and 18 months were 96.0% (95% CI, 91.7–100) and 92.6% (95% CI, 85.1–100) versus 96.6% (95% CI, 94.7–98.4) and 90.2% (95% CI, 86.7–93.9), respectively (*p* = 0.85, log-rank test) (Appendix A). Median OS was not reached for either group at the time of this report.

### 3.6. Propensity Score Matching

Our propensity score matching procedures produced comparable patient cohorts, as shown by the SMD, and showed similar results regarding RFS and OS. Only two BRAF/MEK-treated patients could not be matched to anti-PD-1-treated patients using the nearest neighbor matching procedure. For matching purposes, a 1:1 ratio was chosen to maximize similarities between patient cohorts. Unmatched patients showed no substantial difference compared to matched patients for both BRAF/MEK- and anti-PD-1-treated cohorts.

#### 3.6.1. Nearest Neighbor with Caliper Matching

After 1:1 nearest neighbor with caliper propensity score matching, 112 BRAF/MEK-treated patients were matched with 112 anti-PD-1 patients. Groups did not differ regarding age, sex, ECOG PS, tumor stage, or comorbidities (Table 1). In total, 422 patients were not matched: 2 were treated with BRAF/MEK inhibitors and 420 with anti-PD-1 treatment. Patient characteristics for unmatched patients are shown in Appendix A.

##### Matched BRAF/MEK- and Anti-PD-1-Treated Patients Using Nearest Neighbor with Caliper Propensity Score Matching

With the currently limited follow-up, recurrence-free survival (RFS) in BRAF/MEK-treated patients was superior to matched anti-PD-1-treated patients (*p* = 0.03, stratified log-rank test). RFS rates at 12 and 18 months were 84.4% (95% CI, 76.4–93.1) and 69.7% (95% CI, 57.3–84.9) compared to 68.0% (95% CI, 59.2–78.2) and 68.0% (95% CI, 59.2–78.2), respectively (Figure 3). The predicted median RFS in BRAF/MEK-treated patients was 30.1 months (95% CI, 24.1-NR) and had yet to be reached in matched anti-PD-1-treated patients. However, patients treated with adjuvant BRAF/MEK-inhibition therapy did not have a significantly reduced hazard for recurrence or death compared to matched anti-PD-1-treated patients (HR with a robust variance estimator of 0.64 (95% CI, 0.38–1.10), *p* = 0.11).

Overall survival (OS) in BRAF/MEK-inhibitor-treated patients and matched anti-PD-1-treated patients did not differ at 12 and 18 months: 96.0% (95% CI, 91.6–100.0) and 92.5% (95% CI, 85.0–100) compared to 97.6% (95% CI, 94.4–100) and 94.2% (95% CI, 88.6–100), respectively (*p* = 0.40, stratified log-rank test) (Figure 4). Median OS was not reached for either cohort at the time of this report.

#### 3.6.2. Optimal Matching

After optimal matching, all 114 BRAF/MEK-treated patients were matched with an anti-PD-1-treated patient. Matched patients did not differ regarding age, sex, ECOG PS, the presence of comorbidities, and AJCC tumor stage. The patient characteristics of matched patients after optimal matching are shown in Appendix A.

Recurrence-free survival (RFS) was not significantly different between matched BRAF/MEK-inhibitor-treated patients and anti-PD-1-treated patients (*p* = 0.12, stratified log-rank test) with 12- and 18-month RFS rates of 84.6% (95% CI, 76.8–93.3) and 67.6% (95% CI, 54.9–83.1) compared to 75.8% (95% CI, 67.7–84.9) and 73.1% (95% CI, 63.9–83.6), respectively (Appendix A). The predicted median RFS in BRAF/MEK-treated patients was 30.1 months (95% CI, 20.8-NR) and 35.5 months (95% CI, 24.4-NR) in matched anti-PD-1-treated patients. Patients treated with adjuvant BRAF/MEK-inhibition therapy did not have a significantly reduced hazard of recurrence or death compared to matched anti-PD-1-treated patients (HR with a robust variance estimator of 0.86 (95% CI, 0.49–1.51), *p* = 0.60).

Overall survival (OS) in BRAF/MEK- and matched anti-PD-1-treated patients did not differ at 12 and 18 months: 96.0% (95% CI, 91.7–100.0) and 92.6% (95% CI, 85.1–100) compared to 96.4% (95% CI, 92.5–100) and 92.0% (95% CI, 85.1–99.5), respectively (*p* = 0.70, stratified log-rank test) (Appendix A). Median OS was not yet reached in both patient groups.

## 4. Discussion

Our data show that in daily clinical practice, comorbidities and geographical region are significantly associated with receiving adjuvant BRAF/MEK-inhibition therapy or adjuvant anti-PD-1 systemic treatment in patients with resectable stage III melanoma. We report a better RFS rate at 12 months but a comparable RFS rate at 18 months in BRAF/MEK-treated patients compared to matched anti-PD-1-treated patients using the nearest neighbor matching method. OS did not differ between matched patient populations. Optimal matching, however, showed no significant differences between RFS and OS. In daily clinical practice, 11.5% of patients treated with adjuvant BRAF/MEK-inhibition therapy experience grade ≥ 3 toxicity, which was the most common cause of premature treatment discontinuation. To our knowledge, this is the first report directly comparing adjuvant BRAF/MEK-inhibition therapy to adjuvant anti-PD-1 treatment in daily clinical practice. These results offer insight into the adjuvant treatment with BRAF/MEK-inhibition therapy beyond the clinical trial setting.

### 4.1. Predictors for BRAF/MEK-Inhibition Therapy as First-Line Adjuvant Treatment

In the adjuvant treatment setting, head-to-head comparisons of targeted therapy (TT) and immune checkpoint inhibitors (ICI) have not been made in clinical trials [2,37,38], leaving the decision for first-line treatment to patient and treating physician’s preference. In this report, we show that the presence of comorbidities and geographical region guide the decision for adjuvant systemic therapy of choice. Notably, the delayed availability of adjuvant TT compared to ICI in the Netherlands also accounted for lesser adjuvant TT use until November 2020. We show that patients treated with BRAF/MEK-inhibition therapy significantly more often had autoimmune diseases than anti-PD-1-treated patients and that age is not associated with the choice for TT or ICI.

These findings partially align with the conclusions of Lodde et al. They studied factors influencing adjuvant therapy decisions and the decision for ICI versus TT using real-world data using log-binominal regression analysis [39]. In line with our results, the authors demonstrated a preference for TT in stage III BRAF-mutant melanoma patients with autoimmune diseases (76.5% versus 23.5%, *p* = 0.02). Lodde et al. also described a significant effect of geographical region on the first-line adjuvant treatment choice. However, unlike our data, they did not find any relevant differences in treatment preferences in patients with comorbidities (modified Charleston Comorbidity Index (CCI)).

### 4.2. BRAF/MEK-Inhibition Therapy Toxicity, Premature Treatment Cessation, and Subsequent Treatment

In our previous report on adjuvant anti-PD-1 therapy in daily clinical practice, we reported a slightly higher toxicity rate than the registration trials [12]. In the current study, we report strikingly lower rates of grade ≥ 3 adverse events in BRAF/MEK-treated patients compared to the COMBI-AD trial (11.5% versus 41% in the COMBI-AD trial) [1]. However, we report similar treatment discontinuation rates due to any grade toxicity in BRAF/MEK-treated patients in daily clinical practice compared to trial patients (33% versus 26% in the COMBI-AD trial) [1]. A recent follow-up study to the Lodde et al. study by Livingstone et al. reported similar real-world results to our present study. In their study, the authors showed slightly higher premature discontinuation rates (44%) in real-world patients treated with TT and showed that 60% of patients who prematurely discontinued treatment had done so due to toxicity [40]. The discrepancy in grade ≥3 adverse events rates and treatment discontinuation rates due to any grade adverse events might be due to the under-reporting of these events in daily clinical practice. This under-reporting might be because grade ≥3 adverse events in patients treated with TT is less consequential than grade ≥3 adverse events in ICI-treated patients in which grade ≥3 adverse events usually prompt treatment discontinuation and immunosuppressive treatment.

### 4.3. Recurrence-Free Survival in BRAF/MEK-Treated Patients and Matched Anti-PD-1-Treated Patients

To our knowledge, this is the first study comparing RFS and OS in matched adjuvant-treated BRAF/MEK and anti-PD-1 patients in daily clinical practice. We report a significantly better 1-year RFS for matched adjuvant BRAF/MEK-treated patients compared to adjuvant anti-PD-1-treated patients using nearest neighbor matching but comparable RFS at 18 months. BRAF/MEK-inhibition therapy did not significantly reduce the hazard of recurrence or death compared to treatment with anti-PD therapy in our patient population. The RFS and OS rates reported for BRAF/MEK-treated patients in this study are similar to the RFS and OS rates reported in the registration trials [15,37,38]. Livingstone et al. described a recurrence rate of 35% in TT and 48% in ICI at a median follow-up time of 25.3 and 24.6 months, respectively, in resected stage III/IV melanoma patients registered into a central German registry [40]. Unfortunately, our data are based on shorter follow-up times, as BRAF/MEK-inhibition therapy was approved and reimbursed later in the Netherlands compared to Germany. Further follow-up is necessary to compare these real-world recurrence rates.

### 4.4. BRAF/MEK-Inhibitor or Anti-PD-1 Therapy?

BRAF/MEK inhibitors and anti-PD-1 therapy risks and benefits should be assessed when considering first-line treatment in adjuvant melanoma patients. While the toxicity of BRAF/MEK inhibitors can have a severe impact on quality of life, it generally subsides once treatment is stopped. In contrast, anti-PD1 therapy can cause irreversible toxicity [41]. Moreover, although we previously showed that ICI efficacy in advanced melanoma patients with autoimmune disease (AID) was not inferior [42], ICI is usually avoided in patients with AID in the adjuvant setting, especially when there is the alternative option of BRAF/MEK-inhibition. This is illustrated by the relative low number of patients with AID in our anti-PD1-treated cohort (Appendix A).

With our current limited follow-up, the RFS after one year appears favorable for BRAF/MEK-inhibitor-treated patients compared to anti-PD-1-treated patients. However, we observed a steady drop in RFS throughout the follow-up period in our BRAF/MEK patient population compared to a seemingly emerging plateau in our anti-PD-1 population. As a result, these survival curves appear to cross at approximately 24 months. This is in line with the pattern seen in the cross-study comparison of data from the registration trials [43] and the recent DREAMseq and secombit trial in the advanced setting [44,45], showing better survival for first-line ICI-treated patients in the long run.

### 4.5. Strengths, Limitations, and Future Research

The DMTR database is a prospective nationwide quality registry facilitated by the Dutch Institute for Clinical Auditing (DICA) [11]. Data in the DMTR are well registered and have a high level of completeness [12]. A limitation of this study is its observational nature. In this study, we could only match patients for a limited number of covariates. Thus, the two patient cohorts may differ more than appears in this paper. Lodde et al. describe the fear of adverse events as one of the most common reasons for opting out of adjuvant therapy, regardless of specific therapy choice [39]. Further research should be conducted into patient and physician considerations when choosing TT or ICI as first-line adjuvant therapy. Although propensity score matching mimics a clinical trial setting, an RCT is needed to confirm the results of this study to correct for any residual confounding biases that remain after PSM.

Furthermore, a limitation of this study is that (partly) due to the limited follow-up, an analysis of the second-line treatment and its effect on OS was not feasible. More extended follow-up and a higher number of patients will more reliably indicate the comparative effectiveness of adjuvant BRAF/MEK-inhibition and anti-PD-1 therapy.

## 5. Conclusions

Comorbidities and geographical region play a role in deciding the adjuvant treatment of choice in resected stage III BRAF-mutant melanoma patients in the Netherlands. In patients with resected stage III BRAF-mutant melanoma, BRAF/MEK-inhibition resulted in a better 12-month RFS than anti-PD-1 in matched patients, but this difference was no longer observed at 18 months. Therefore, a longer follow-up is necessary to confirm these results and estimate the long-term efficacy of these adjuvant treatments in daily clinical practice.

## Figures and Tables

**Figure 1 cancers-15-00409-f001:**
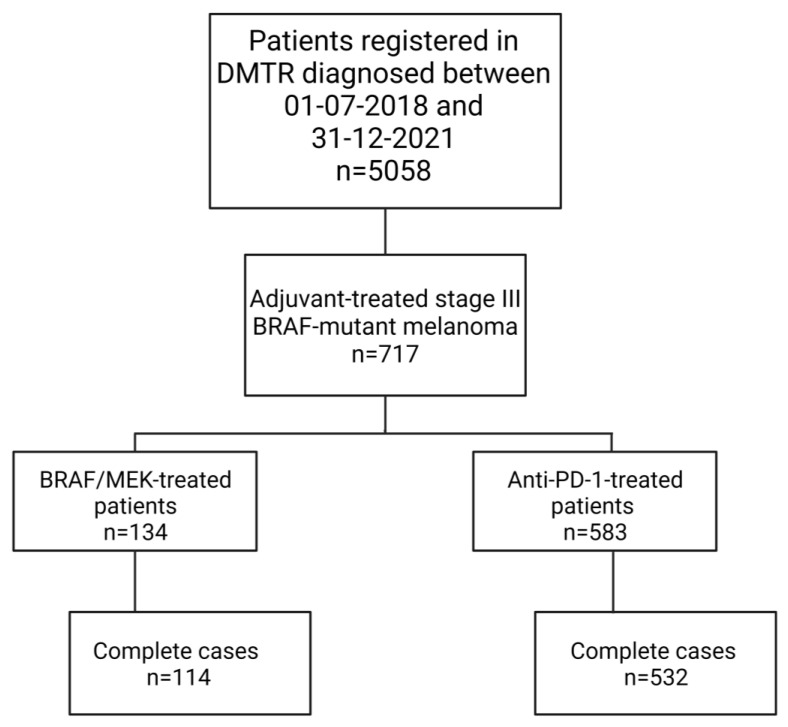
Flowchart study of the population. Created with BioRender.com.

**Figure 2 cancers-15-00409-f002:**
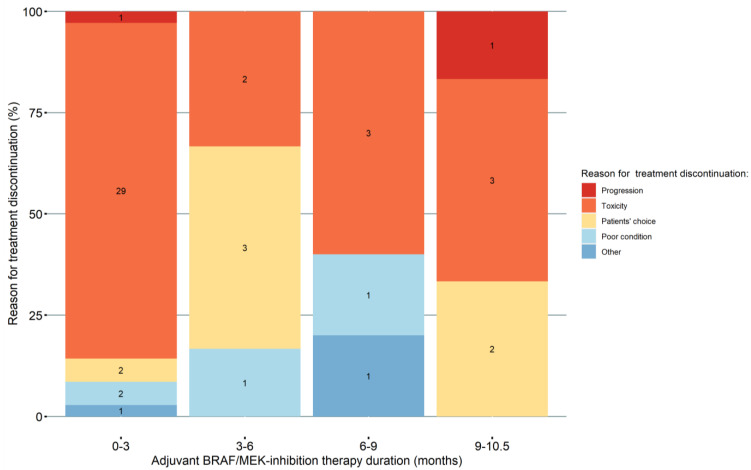
Timing and reason for premature treatment discontinuation of adjuvant BRAF/MEK-inhibition therapy patients. This figure depicts all patients that discontinued BRAF/MEK-treatment (n = 52). Adjuvant BRAF/MEK-inhibition therapy duration is depicted as time from starting therapy to last prescription date.

**Figure 3 cancers-15-00409-f003:**
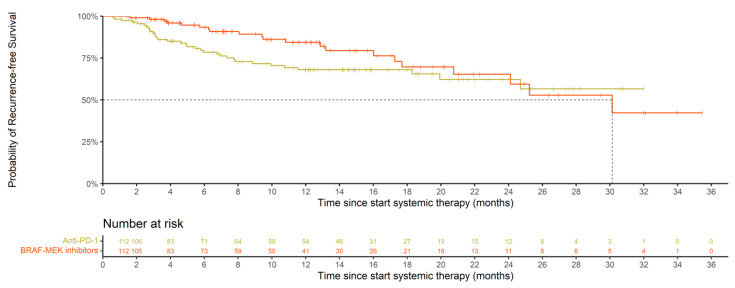
Recurrence-free survival (RFS) of BRAF/MEK- and anti-PD-1-treated patients after nearest neighbor propensity score matching.

**Figure 4 cancers-15-00409-f004:**
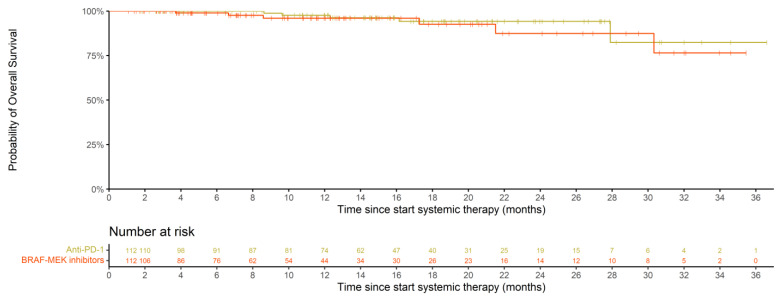
Overall survival (OS) of BRAF/MEK- and anti-PD-1-treated patients after nearest neighbor propensity score matching.

**Table 1 cancers-15-00409-t001:** Patient and tumor characteristics of cutaneous melanoma patients treated with adjuvant BRAF/MEK-inhibition therapy and adjuvant anti-PD-1-treated patients.

	Original Sample					Nearest Neighbor Matching			
		BRAF/MEK-Inhibition Therapy	Anti-PD-1 Therapy	*p*-Value	SMD	BRAF/MEK-Inhibition Therapy	Anti-PD-1 Therapy	*p*-Value	SMD
**n (%)**		114 (17.6)	532 (82.4)	<0.01		112	112	1.000	<0.01
**Age**	<65	76 (66.7)	345 (64.8)	0.80	0.04	76 (67.9)	75 (67.0)	1.000	0.02
	≥65	38 (33.3)	187 (35.2)			36 (32.1)	37 (33.0)		
**Sex**	1	63 (55.3)	313 (58.8)	0.55	0.07	63 (56.2)	63 (56.2)	1.000	<0.01
	2	51 (44.7)	219 (41.2)			49 (43.8)	49 (43.8)		
**ECOG PS**	0	82 (71.9)	399 (75.0)	0.57	0.07	80 (71.4)	81 (72.3)	1.000	0.02
	≥1	32 (28.1)	133 (25.0)			32 (28.6)	31 (27.7)		
**Comorbidities**	No	29 (25.4)	194 (36.5)	0.03	0.24	27 (24.1)	26 (23.2)	1.000	0.02
	Yes	85 (74.6)	338 (63.5)			85 (75.9)	86 (76.8)		
**AJCC Tumor Stage**	IIIA	18 (15.8)	53 (10.0)	0.09	0.26	18 (16.1)	17 (15.2)	1.000	0.03
	IIIB	31 (27.2)	200 (37.6)			31 (27.7)	31 (27.7)		
	IIIC/D	51 (44.7)	227 (42.7)			49 (43.8)	49 (43.8)		
	Unknown	14 (12.3)	52 (9.8)			14 (12.5)	15 (13.4)		

**Table 2 cancers-15-00409-t002:** Multivariable logistic regression for receiving BRAF/MEK inhibitors in study population.

	OR	95% CI	*p*-Value
**(Intercept)**	0.09	0.05–0.18	<0.01
**Age**			
<65	Ref.		
≥65	0.73	0.45–1.16	0.19
**Sex**			
Male	Ref.		
Female	1.26	0.82–1.92	0.29
**ECOG PS**			
0	Ref.		
>=1	1.2	0.73–1.93	0.46
**Comorbidities**			
No	Ref.		
Yes	1.67	1.03–2.76	0.04
**AJCC 8th edition tumor stage**			
IIIC/IID	Ref.		
IIIA	1.46	0.76–2.72	0.24
IIIB	0.70	0.43–1.15	0.17
Unknown	1.05	0.52–2.04	0.88
**Geographical region**			
Middle region	Ref.		
Northern region	3.05	1.68–5.63	<0.01
Southern region	1.6	0.95–2.75	0.08

Ref.—reference.

**Table 3 cancers-15-00409-t003:** Toxicity rates (grade ≥ 3) of adjuvant BRAF/MEK-treated stage III/IV melanoma patients in daily clinical practice (n = 104).

Toxicity Type	Toxicity Rates (%)
Abnormal laboratory values	1.9
Arthralgia	1.0
Liver failure	1.0
Malaise/dizziness	1.0
Neuropathy	1.0
Pyrexia	4.8
Skin toxicities	3.8
Visual changes/retinopathy/occlusion of retinal vein	1.0
Other/unknown	2.9

## Data Availability

The data presented in this study are available on request from the corresponding author. The data are not publicly available due to Dutch privacy laws.

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
