# Peer review of "Adjuvant BRAF-MEK Inhibitors versus Anti PD-1 Therapy in Stage III Melanoma: A Propensity-Matched Outcome Analysis"

_cancers, 2023, doi:10.3390/cancers15020409_

Round 1

Reviewer 1 Report

In this study, the authors compared the clinical impact of adjuvant BRAF-MEK inhibitors vs. anti PD-1 therapy in stage 3 melanoma patients. The results showed that geographical region and comorbidities appear correlated to adjuvant therapy selection, and that the recurrence-free survival were significantly different at 12 months but the significance vanished at later timepoints. The study overall can be of interest for researchers in the field.

In line278 the number of patients selected for each group is 112, which does not match the number 110 in Table 1.

The authors only mentioned the premature discontinuation and toxicity of adjuvant BRAF/MEK inhibition therapy, but did not mention these for the anti PD-1 treatment group patients.

The authors mentioned (such as at line196) that a subset of patients have BRAF non-V600E mutation. Does this mean other BRAF V600 codon mutations, or any other BRAF mutations (other than V600E)? Please clarify.

There are many obvious typos in the manuscript. For example, in line72 "in" should be "is", line92 "t" should be "at", line 163 "abovementioned" should be "above-mentioned". Please carefully review the manuscript and correct these and other possible typos.

Please remove the title "Propensity score matching" (line 98) in background.

Author Response

Reviewer 1

In this study, the authors compared the clinical impact of adjuvant BRAF-MEK inhibitors vs. anti PD-1 therapy in stage 3 melanoma patients. The results showed that geographical region and comorbidities appear correlated to adjuvant therapy selection, and that the recurrence-free survival were significantly different at 12 months but the significance vanished at later timepoints. The study overall can be of interest for researchers in the field.

In line 278 the number of patients selected for each group is 112, which does not match the number 110 in Table 1.

Response: We thank the reviewer for pointing this out. The numbers in the table have been corrected.

The authors only mentioned the premature discontinuation and toxicity of adjuvant BRAF/MEK inhibition therapy, but did not mention these for the anti PD-1 treatment group patients.

Response: We have previously published our findings on premature discontinuation and toxicity rates of adjuvant anti-PD-1 therapy in the European Journal of Cancer [1]. We have added these findings to the current manuscript (lines 226-228 and lines 246-247).

The authors mentioned (such as at line 196) that a subset of patients have BRAF non-V600E mutation. Does this mean other BRAF V600 codon mutations, or any other BRAF mutations (other than V600E)? Please clarify.

Response: The BRAF non-V600E mutation group consists of patients with V600E, V600K, V600R and “other” or unknown BRAF mutations. We have updated the manuscript and supplement to include more detailed BRAF mutation data.

There are many obvious typos in the manuscript. For example, in line72 "in" should be "is", line92 "t" should be "at", line 163 "abovementioned" should be "above-mentioned". Please carefully review the manuscript and correct these and other possible typos.

Response: We thank the reviewer for pointing this out. The typos have been corrected.

Please remove the title "Propensity score matching" (line 98) in background.

Response: I have removed the title as suggested.

Reviewer 2 Report

This is an interesting study to evaluate the effects of BRAF targeted therapy in the adjuvant space.  The authors utilize a propensity score matching to "match" as best as possible outside of a trial BRAF treated patients to anti-PD1 treated patients.  Given that we are not likely to see this trial in practice, this is a reasonable design.  

1. It is important to tabulate OS in this setting.  It is understood that there will be a lot of censored data, but in the metastatic setting immunotherapy is used first in many circumstances due to the results of the DreamSeq trial.  The question is whether the patients do very poorly after relapse of BRAF therapy.  Either way, this is extremely valuable information.  A relative short follow up period is needed for this (~6 months) and it appears that the authors should have access to this information.   The other way to approach this is to state these limitations with regard to overall survival.  This can be tackled in a new section in the Discussion.  

2. In Figure 2, what is meant by poor condition vs discontinuation from toxicity.  It would also be useful to summarize the "others" in a de-identified way if possible.  

3. Figure 3:  Please increase the font size.  In general, please increase the font size in the figures.  

4. The title in the discussion should be "Relapse Free Survival in BRAF/MEK-treated....    

Author Response

Reviewer 2

This is an interesting study to evaluate the effects of BRAF targeted therapy in the adjuvant space.  The authors utilize a propensity score matching to "match" as best as possible outside of a trial BRAF treated patients to anti-PD1 treated patients.  Given that we are not likely to see this trial in practice, this is a reasonable design.  

  1. It is important to tabulate OS in this setting. It is understood that there will be a lot of censored data, but in the metastatic setting immunotherapy is used first in many circumstances due to the results of the DreamSeq trial. The question is whether the patients do very poorly after relapse of BRAF therapy. Either way, this is extremely valuable information. A relative short follow up period is needed for this (~6 months) and it appears that the authors should have access to this information. The other way to approach this is to state these limitations with regard to overall survival. This can be tackled in a new section in the Discussion.  

Response: We thank the reviewer for this valuable suggestion. Due to the relative short follow-up, data on second-line treatment were unfortunately too limited to reliably draw conclusions. We will address this important topic in a subsequent analysis with more follow-up. We have added this to our limitations section, as advised (line 411).

  1. In Figure 2, what is meant by poor condition vs discontinuation from toxicity.  It would also be useful to summarize the "others" in a de-identified way if possible.

Response: We thank the reviewer for pointing out this unclarity. In figure 2, “poor condition” refers to patients with a poor/deteriorated condition (due to any reason; this can be a combination of toxicity, comorbidity and/or disease progression and other factors), which does not allow for BRAF/MEK-inhibition therapy. “Discontinuation due to toxicity” refers to patients previously fit enough for BRAF/MEK-inhibition therapy who have developed toxicity that is severe enough to warrant discontinuation of further treatment.

The “other” category is not specifically defined in the DMTR database, so unfortunately it is not possible to further specify this group.

  1. Figure 3:  Please increase the font size.  In general, please increase the font size in the figures. 

Response: The font size has been increased in the figures.

  1. The title in the discussion should be "Relapse Free Survival in BRAF/MEK-treated....

Response: The title has been corrected in the manuscript.

Round 2

Reviewer 2 Report

Thank you to the authors for providing a sentence on the lack of ability of performing OS survival analysis at this time.  

The effect dissipates at 18 months.  Therefore, the OS survival analysis in a subsequent paper will be eagerly awaited by the community.  This paper is as close as we are going to get to a targeted vs immune therapy study in the adjuvant setting

Recommendation:  One paragraph is needed to explain the complex medical decision process in the discussion. It is obvious to the authors, but it might not be obvious to the general audience that even though BRAF targeted has a longer RFS that anti-PD1 based therapy may have a longer OS in the long run.   Would recommend to place this into context of the recent immune therapy vs targeted therapy trials in the metastatic setting.  Some comment on patients with autoimmunity (psoriasis, rheumatoid arthritis, ulcerative colitis - whichever patients you have in your cohort).   If you cannot make any comments regarding autoimmune conditions, then say so.  

Author Response

Reviewer 2

Thank you to the authors for providing a sentence on the lack of ability of performing OS survival analysis at this time. 

The effect dissipates at 18 months. Therefore, the OS survival analysis in a subsequent paper will be eagerly awaited by the community. This paper is as close as we are going to get to a targeted vs immune therapy study in the adjuvant setting

Recommendation:  One paragraph is needed to explain the complex medical decision process in the discussion. It is obvious to the authors, but it might not be obvious to the general audience that even though BRAF targeted has a longer RFS that anti-PD1 based therapy may have a longer OS in the long run. Would recommend to place this into context of the recent immune therapy vs targeted therapy trials in the metastatic setting. Some comment on patients with autoimmunity (psoriasis, rheumatoid arthritis, ulcerative colitis - whichever patients you have in your cohort).   If you cannot make any comments regarding autoimmune conditions, then say so.

Response:

We thank the reviewer for this valuable suggestion. Accordingly, we have added a paragraph to the discussion in the manuscript addressing the abovementioned points.

“BRAF/MEK-inhibitor or anti-PD-1 therapy?

BRAF/MEK inhibitors and anti-PD-1 therapy risks and benefits should be assessed when considering first-line treatment in adjuvant melanoma patients. While toxicity of BRAF/MEK inhibitors can have severe impact on quality of life, it generally subsides once treatment is stopped. In contrast, anti-PD1 therapy can cause irreversible toxicity [41]. Moreover, although we previously showed that ICI efficacy in advanced melanoma patients with autoimmune disease (AID) was not inferior [42], ICI is usually avoided in patients with AID in the adjuvant setting, especially when there is the alternative option of BRAF/MEK-inhibition. This is illustrated by the relative low number of patients with AID in our anti-PD1 treated cohort (supplementary table 2).

With our current limited follow-up, the RFS after one year appears favorable for BRAF/MEK inhibitor compared to anti-PD-1 treated patients. However, we observed a steady drop in RFS throughout the follow-up period in our BRAF/MEK-patient population compared to a seemingly emerging plateau in our anti-PD-1 population. As a result, these survival curves appear to cross at approximately 24 months. This is in line with the pattern seen in the cross-study comparison of data from the registration trials [43] and the recent DREAMseq and secombit trial in the advanced setting [44, 45], showing better survival for first-line ICI treated patients in the long run.”

Round 3

Reviewer 2 Report

The language now appropriately addresses the complex nature of this clinical problem.